# A Mixed-Integer and Asynchronous Level Decomposition with Application to the Stochastic Hydrothermal Unit-Commitment Problem

**Bruno Colonetti [1,\*], Erlon Cristian Finardi [1,2]**  **and Welington de Oliveira [3,\*]**

[1] Department of Electrical and Electronic Engineering, Federal University of Santa Catarina, Florianópolis 88040-900, Brazil; erlon.finardi@ufsc.br
[2] INESC P&D Brasil, Bairro Gonzaga 11055-300, Brazil
[3] MINES ParisTech, CMA—Centre de Mathématiques Appliquées, PSL—Research University, Sophia Antipolis, 75006 Paris, France
\* Correspondence: colonetti.bruno@posgrad.ufsc.br (B.C.); welington.oliveira@mines-paristech.fr (W.d.O.)

**Abstract:** Independent System Operators (ISOs) worldwide face the ever-increasing challenge of coping with uncertainties, which requires sophisticated algorithms for solving unit-commitment (UC) problems of increasing complexity in less-and-less time. Hence, decomposition methods are appealing options to produce easier-to-handle problems that can hopefully return good solutions at reasonable times. When applied to two-stage stochastic models, decomposition often yields subproblems that are embarrassingly parallel. Synchronous parallel-computing techniques are applied to the decomposable subproblem and frequently result in considerable time savings. However, due to the inherent run-time differences amongst the subproblem's optimization models, unequal equipment, and communication overheads, synchronous approaches may underuse the computing resources. Consequently, asynchronous computing constitutes a natural enhancement to existing methods. In this work, we propose a novel extension of the asynchronous level decomposition to solve stochastic hydrothermal UC problems with mixed-integer variables in the first stage. In addition, we combine this novel method with an efficient task allocation to yield an innovative algorithm that far outperforms the current state-of-the-art. We provide convergence analysis of our proposal and assess its computational performance on a testbed consisting of 54 problems from a 46-bus system. Results show that our asynchronous algorithm outperforms its synchronous counterpart in terms of wall-clock computing time in 40% of the problems, providing time savings averaging about 45%, while also reducing the standard deviation of running times over the testbed in the order of 25%.

**Keywords:** stochastic programming; stochastic hydrothermal UC problem; parallel computing; asynchronous computing; level decomposition

## 1. Introduction

The unit-commitment (UC) problem aims at determining the optimal scheduling of generating units to minimize costs or maximize revenues while satisfying local and system-wide constraints [1]. In its deterministic form, UC still poses a challenge to operators and researchers due to the large sizes of the systems and the increasing modeling details necessary to represent the system operation. For instance, in the Brazilian case, the current practice is to set a limit of 2 h for the solution of the deterministic UC [2], while the Midcontinent Independent System Operator (MISO) sets a time limit of 20 min for its UC [3]. (Note that the Brazilian system and the MISO are different from a physical, as well as from a market-based, viewpoint, but the problems being solved in these two cases share the same classical concept of the UC.) Nonetheless, the growing presence of intermittent generation has

added yet more difficulty to the problem, giving rise to what is called uncertain UC [4]. The latter is considerably harder to solve than its deterministic counterpart, and one of the reasons for its lack of adoption in the industry is precisely its computational burden: Large-scale uncertain UC takes a prohibitively long time to be solved. In this context, efficient solution methods for the uncertain UC that can take full advantage of the computational resources at hand are both desirable and necessary to help system operators cope with uncertain resources.

In particular, to model the uncertainty arising from renewable sources, one of two approaches is generally employed: robust optimization or stochastic programming [4]. The latter is by far the most employed, both in its chance-constrained and recourse variants. In stochastic programs with recourse, uncertainty is, in general, represented by finite-many scenarios, and the problem is formulated either in a two-stage or multistage setting. In two-stage stochastic problems, the first-stage variables must be decided before uncertainty is revealed. Once the uncertain information becomes known, recourse actions are taken to best accommodate the first-stage decisions [5]. In stochastic hydrothermal unit-commitment (SHTUC) problems, the sources of uncertainties are related to renewable resources, spot prices, load, and equipment availability [1,4].

The commitment decisions are usually modeled as first-stage variables, while dispatch decisions are the recourse actions (second-stage variables). Given the mixed-integer nature of commitment decisions, SHTUC problems in a two-stage formulation give rise to large-scale mixed-integer optimization models whose numerical solution by off-the-shelf solvers is often prohibitive due to time requirements or limited computing resources. Consequently, decomposition techniques must come into play [1,4,6,7]. Benders decomposition (BD) and Lagrangian relaxation (LR) are the most used techniques to handle SHTUC problems. While the BD deals with the primal problem [8], LR is a dual procedure employed to compute the best lower bound for the SHTUC problem [7,9]. Primal-recovery heuristics are employed to compute primal-feasible points, which are not, in general, optimal solutions. This is the main shortcoming of LR-based techniques.

Decomposition techniques yield models that are amenable for parallelization [5]. A common strategy for solving problems simultaneously is to use a master/worker framework with pre-specified synchronization points [10], which we call synchronous computing (SYN). In this framework, the master chooses new iterates and sends them to workers, who, in turn, are responsible for solving one or more subproblems. Examples of SYN implementations for UC are given in [11–14]. An aspect of SYN is that, at predetermined points of the algorithm, the master must wait for all workers to respond to resume the iterative process: the synchronization points. However, the times for workers to finish their respective tasks might vary significantly. This results in idle times, both for the master and for workers who respond quickly [10]. One way to reduce idle times is to use asynchronous computing (ASYN).

In contrast to SYN, in ASYN, there are no synchronization points, so the master and workers do not need to wait until all workers respond to continue their operations. Thus, in an iterative process, e.g., in BD, the master would compute the next iterate based on information of possibly only a proper, but nonempty, subset of the workers. Based on this possibly incomplete information, the master sends a new iterate to available workers, while slower workers are still carrying their tasks on an outdated iterate. Because in ASYN iterates might not be evaluated by all workers, the evaluation of the objective function (yielding bounds on the optimal values) is precluded. Hence, a fundamental step in ASYN is the (scarce) coordination of workers to produce valid bounds.

ASYN implementations have been proposed in the UC literature mainly to solve the dual problems (issued by LRs) via either subgradient algorithms or cutting-plane-based methods [15–17]. In References [15,16], a queue of iterates is created and its elements are gradually sent to the workers. Auxiliary lists keep track of the evaluation status of each worker with respect to the elements in the queue. Once an element has been evaluated by all workers, a valid bound to the original problem is available. The authors of Reference [15] demonstrate that their algorithm converges to a dual global solution regardless of the iterate-selection policy used to choose the iterates from the queue—first-in-first-out or last-in-first-out. In References [17], the authors keep a list of all the iterates

to compute valid bounds. In addition to solving the dual problem asynchronously, Reference [17] also conducts the primal recovery asynchronously. While References [15,16] employ a convex trust-region bundle method, Reference [17] implements an incremental subgradient method. Asynchronous implementations of BD for convex problems can be found in References [18–20]. In Reference [18], the dual dynamic-programming algorithm is handled asynchronously in a hydrothermal scheduling problem. In Reference [19], the stochastic dual dynamic-programming algorithm is used for addressing the long-term planning problem of a hydro-dominated system: The authors propose to compute Benders cuts in an asynchronous fashion. This is also the case in Reference [20], where the authors consider an asynchronous Benders decomposition for convex multistage stochastic programming.

Despite being successfully applied in a variety of fields, e.g., References [18,19] and the references in References [21], the classical BD is well-known to suffer from slow convergence due to the oscillatory nature of Kelley's cutting-plane method [22,23]. Regularized BDs have been proven to outperform the classical one in several problems: See Reference [24] for (convex) two-stage linear programming, Reference [25] for (nonconvex) chance-constrained problems, and Reference [26] for robust designed of stations in water distribution networks. Several types of regularization exist [25,27,28]: proximal, trust-region, and level sets. Among the regularization methods, the level bundle method [29], also known as level decomposition (LD) in two-stage programming [24], stands out for its flexibility in dealing with convex or nonconvex feasible sets, stability functions and centers, and inexact oracles [25,26,30]. Recently, asymptotically level bundle methods for convex optimization were proposed in Reference [31]. The paper presents two algorithms. The first one does not employ coordination, but it makes use of upper bounds on the Lipschitz constants of the involved functions to compute upper bounds for the problem. The second algorithm does not make use of the latter assumption but requires scarce coordination. The authors of Reference [31] focus on the convergence analysis of their proposals (suitable only for the convex setting) and present limited numerical experiments. In this work, we build on Reference [31] and extend its asynchronous algorithm with scarce coordination (Algorithm 3 of Reference [31]) to the mixed-integer setting. Moreover, we consider a more general setting in which tasks can be assigned to works in a dynamic fashion, as described in Section 3. We highlight that the convergence analysis given in Reference [31] relies strongly on elements of convex analysis such as the Smulian's theorem and the Painlevé–Kuratowski set convergence. Such key theoretical results are no longer valid in the setting of nonconvex sets, and hence the convergence analysis developed in Reference [31] does not apply to our mixed-integer setting. For this reason, the convergence analysis of our asynchronous LD must be done anew. We not only provide convergence analysis of our method but also assess its numerical performance on a test set consisting of 54 instances of two-stage UC problems with mixed-integer variables in the first stage.

We care to mention that other asynchronous bundle methods exist in the literature, but they are all designed for convex optimization problems [15,16,32]. The latter reference proposes an asynchronous proximal bundle method, whereas References [15,16] consider a trust-region variant for polyhedral functions. Our approach, which follows the lines of the extended level bundle method of Reference [30], does not require the involved functions to be polyhedral or the feasible set to be convex. As an additional advantage, our algorithm is easily implementable.

This work is organized as follows. Section 2 presents a generic formulation of our two-stage SHTUC problem. The extended asynchronous LD and its convergence analysis are presented in Sections 2.1 and 2.2, respectively. Section 3 presents more details of the considered SHTUC problem and states our case studies. Numerical experiments assessing the benefits of our proposal are given in Section 4. Finally, in Section 5, we present our final remarks.

## 2. Materials and Methods

We address the problem of an Independent System Operator (ISO) in a hydro-dominated system with a loose-pool market framework. The ISO decides the day-ahead commitment considering operation costs, forecast errors in wind generation, and inflows; and the usual generation and system-wide

constraints. The uncertainties in wind and inflows are represented by a finite set of scenarios, $\mathcal{S}$, and the decisions are made in two stages. At the first stage, the ISO decides on the commitment of units, whereas, at the second stage, the operator determines the dispatch according to the random-variable realization. Full details on the considered stochastic hydrothermal unit-commitment (SHTUC) are given shortly. For presenting our approach, which is not limited to (stochastic) unit-commitment (UC) problems, we adopt the following generic formulation.

$$f_* := \min_{x,y} \left\{ \mathbf{c}^\mathsf{T} x + \sum_{\mathsf{s} \in \mathcal{S}} \mathbf{q}_\mathsf{s}^\mathsf{T} y_\mathsf{s} \,\middle|\, \begin{array}{l} x \in \mathcal{X},\ \mathbf{T}x + \mathbf{W}y_\mathsf{s} \le \mathbf{h}_\mathsf{s}, \\ y_\mathsf{s} \in \mathcal{Y}_\mathsf{s},\ \mathsf{s} \in \mathcal{S} \end{array} \right\}. \tag{1}$$

In this formulation, the n-dimensional vector $x$ represents the first-stage variables with associated cost-vector, $\mathbf{c}$. The second-stage variables, $y_\mathsf{s}$, and their associated costs, $\mathbf{q}_\mathsf{s}$, depend on the scenario, $\mathsf{s} \in \mathcal{S}$. The cost vector, $\mathbf{q}_\mathsf{s}$, is assumed to incorporate the positive probability of scenario s. The first- and second-stage variables are coupled by constraints $\mathbf{T}x + \mathbf{W}y_\mathsf{s} \le \mathbf{h}_\mathsf{s}$: $\mathbf{T}$ is the technology matrix; and $\mathbf{W}$ and $\mathbf{h}_\mathsf{s}$ are, respectively, the recourse matrix and a vector of appropriate dimensions. While $\mathcal{X} \ne \varnothing$ is a compact possibly nonconvex, the scenario-dependent set $\mathcal{Y}_\mathsf{s}$ is a convex polyhedron.

As previously mentioned, depending on the UC problem and number of scenarios, the mixed-integer linear programming (MILP) Problem (1) cannot be solved directly by an off-the-shelf solver. The problem is thus decomposed by making use of the recourse functions.

$$Q_\mathsf{s}(\mathbf{x}) := \min_{y \in \mathcal{Y}_s} \mathbf{q}_\mathsf{s}^\mathsf{T} y \ \text{s.t.}\ \mathbf{W}_\mathsf{s} y \le \mathbf{h}_\mathsf{s} - \mathbf{T}_\mathsf{s} \mathbf{x}. \tag{2}$$

It is well-known that $\mathbf{x} \mapsto Q_\mathsf{s}(\mathbf{x})$ is a non-smooth convex function of $\mathbf{x}$. If the above subproblem has a solution, then a subgradient of $Q_\mathsf{s}$ at $\mathbf{x}$ can be computed by making use of a Lagrange multiplier, $\pi_s$, associated with a constraint, $\mathbf{W}_\mathsf{s} y_\mathsf{s} \le \mathbf{h}_\mathsf{s} - \mathbf{T}_\mathsf{s} \mathbf{x}$: $-\mathbf{T}_\mathsf{s}^\mathsf{T} \pi_s \in \partial Q_\mathsf{s}(\mathbf{x})$. On the other hand, if the recourse function $Q_\mathsf{s}$ is infeasible, then the point $\mathbf{x}$ can be cutoff by adding a feasibility cut [5].

Let $\mathcal{P}$ be a partition of $\mathcal{S}$ into w subsets: $\mathcal{P} = \{P_1, \dots, P_w\}$, with $P_\mathsf{j} \ne \varnothing$ for all $\mathsf{j} \in \{1, \dots, w\}$, and $P_\mathsf{j} \cap P_\mathsf{i} = \varnothing$ for $\mathsf{i} \ne \mathsf{j}$. By defining $f^\mathsf{j}(\mathbf{x}) := \sum_{\mathsf{s} \in P_\mathsf{j}} Q_\mathsf{s}(\mathbf{x})$, Problem (1) can be rewritten as

$$f_* = \min_{x \in \mathcal{X}} \mathbf{c}^\mathsf{T} x + f^1(x) + \dots + f^w(x). \tag{3}$$

In our notation, w stands for the number of workers evaluating the recourse functions. The workers $\mathsf{j} \in \{1, \dots, w\}$ are processes running on a single machine or multiple machines. Likewise, we define a master process—hereafter referred to only as master—to solve the master program (which is defined shortly).

### 2.1. The Mixed-Integer and Asynchronous Level Decomposition

For every point $\mathbf{x}_\mathsf{k}$, where k represents an iteration counter, worker j receives $\mathbf{x}_\mathsf{k}$ and provides us with the first-order information on the component function $f^\mathsf{j}$: the value of the function $f^\mathsf{j}(\mathbf{x}_\mathsf{k})$ and a subgradient [23] $g_\mathsf{k}^\mathsf{j} \in \partial f^\mathsf{j}(\mathbf{x}_\mathsf{k})$, in the two-stage setting, $g_\mathsf{k}^\mathsf{j} := -\sum_{\mathsf{s} \in P_j} \mathbf{T}_\mathsf{s}^\mathsf{T} \pi_\mathsf{s}$. Convexity of $f^\mathsf{j}$ implies that the linearization $f^\mathsf{j}(\mathbf{x}_\mathsf{k}) + \langle g_\mathsf{k}^\mathsf{j}, x - \mathbf{x}_\mathsf{k} \rangle$ approximates $f^\mathsf{j}(x)$ from below for all $x$. By gathering iteration indices into sets $J^\mathsf{j} \subset \{1, 2, \dots, \mathsf{k}\}$ along with the iterations at which $f^\mathsf{j}$ were evaluated, we can construct individual cutting-plane models for functions $f^\mathsf{j}$, with $\mathsf{j} \in \{1, \dots, w\}$: $\min_{\mathsf{i} \in J^\mathsf{j}}\{f^\mathsf{j}(\mathbf{x}_\mathsf{k}) + \langle g_\mathsf{k}^\mathsf{j}, x - \mathbf{x}_\mathsf{k} \rangle\} \le f^\mathsf{j}(x)$. These models define—together with a stability center $\hat{\mathbf{x}}_\mathsf{k}$, a level parameter $f_\mathsf{k}^{\mathsf{lev}} \in \mathfrak{R}$, and a given norm $\|\cdot\|_2$—the following master program (MP)

$$\begin{cases} \min\limits_{x,r} & \|x - \hat{\mathbf{x}}_\mathsf{k}\|_2 \\ \text{s.t.} & \text{possible feasibility cuts} \\ & f^\mathsf{j}(\mathbf{x}_\mathsf{i}) + \langle g_\mathsf{i}^\mathsf{j}, x - \mathbf{x}_\mathsf{i} \rangle \le r_\mathsf{j}, \quad \forall \mathsf{i} \in J_\mathsf{k}^\mathsf{j}, \forall \mathsf{j} = 1, \dots, w \\ & \mathbf{c}^\mathsf{T} x + \sum\limits_{\mathsf{j}=1}^{w} r_\mathsf{j} \le f_\mathsf{k}^{\mathsf{lev}},\ x \in \mathcal{X}. \end{cases} \tag{4}$$

At iteration k, an MP solution is denoted by $\mathbf{x}_{k+1}$. If any $Q_s$ is infeasible at $\mathbf{x}_{k+1}$, then a feasibility cut is added to the MP. We skip further details on this matter, since it is a well-known subject in the literature of two-stage programming [5]. On the other hand, if $\mathbf{x}_{k+1}$ (sent to a work j) is feasible for all recourse functions, $Q_s$, the model $f^j$ in the MP is updated. The improvement in the model $f^j$ is possibly based on outdated iterate $\mathbf{x}_{a(j)}$, where a(j) < k is the iteration index of the *anterior* information provided by worker j. We care to mention that the MP can be infeasible itself depending on the level parameter $f_k^{lev}$. Due to the convexity of the involved functions, if the MP is infeasible, then $f_k^{lev}$ is a valid lower bound, $f_k^{low}$, on $f_*$ [30].

Without coordination, there is no reason for all workers to be called upon the same iterate. This fact precludes the computation of an upper bound, $f_k^{up}$, of $f_*$. Algorithm 2 in Reference [31] deals with this situation without resorting to coordination techniques, but it requires more assumptions on the functions $f^j$: upper bounds on their Lipschitz constants should be known. Since we do not make this assumption, we will need scarce coordination akin to Algorithm 3 of Reference [31] for computing upper bounds on $f_*$. As in Reference [31], the coordination iterates are denoted by $\overline{\mathbf{x}}_k$. Assuming that all workers eventually respond (after an unknown time), the coordination allows them to compute the full value, $f(\overline{\mathbf{x}}_k)$, and a subgradient, $\overline{g} \in \partial f(\overline{\mathbf{x}}_k)$, at the coordination iterate. The function value is used to update the upper bound, $f_k^{up}$, as usual for level methods; the subgradient is used to update the bound L on the Lipschitz constant of $f$.

In our algorithm below, the coordination is implemented by two vectors of Booleans: **to-coordinate** and **coordinating**. The role of **to-coordinate**[j] is to indicate to the master that worker j will evaluate $f^j$ on the new coordination point $\overline{\mathbf{x}}_k$; (at that moment, **to-coordinate**[j] is set to *false*, and **coordinating**[j] is set to *true*). Similarly, **coordinating**[j] indicates to the master that worker j is responding to a coordination step, which is used to update the upper bound. When a worker j responds, it is included in the set $\mathcal{A}$ of available workers. If all workers are busy, then $\mathcal{A} = \varnothing$. Our algorithm mirrors as much as possible Algorithm 3 of Reference [31], but contains some important specificities to handle (i) mixed-integer feasible sets and (ii) extended real-valued objective functions (we do not assume that $f(x)$ is finite for all $x \in \mathcal{X}$). To handle (ii), we furnish our algorithm with a feasibility check (and addition of cuts), and for (i) we not only use a specialized solver for the MP but also change the rule for scarce coordination. The reason is that the rule of Reference [31] is only valid in the convex setting. Under nonconvexity, the coordination test $\|\mathbf{x}_k - \mathbf{x}_{k-1}\| < \frac{\alpha}{L}\Delta_{k-1}$ (with $\alpha \in (0,1)$ and $L \geq \|g_i\|$, $i = 1, \ldots, k$) implies that the following inequality (important for the convergence analysis) is jeopardized:

$$\|\mathbf{x}_k - \hat{\mathbf{x}}_k\|^2 \geq \|\mathbf{x}_{k-1} - \hat{\mathbf{x}}_k\|^2 + \left(\frac{\alpha\Delta_{k-1}}{L}\right)^2. \tag{5}$$

In the algorithm below, coordination is triggered when (5) is not satisfied and all workers have already responded on the last coordination iterate (i.e., rr = 0, where **rr** stands for "remaining to respond").

The assumption that the algorithm starts with a feasible point is made only for the sake of simplicity. Indeed, the initial point can be infeasible, but, in this case, Step 3 must be changed to ensure that the first computed feasible point is a coordination iterate. For the problem of interest, the feasibility check performed at line 45 amounts to verifying if $f(\mathbf{x}_{k+1}) < \infty$. In our SHTUC, the feasibility check comprises an auxiliary problem for verifying if ramp-rate constraints would be violated by $\mathbf{x}_{k+1}$ and an additional auxiliary problem for checking if reservoir-volume bounds would be violated. Both problems are easily reduced to small linear-programming problems that can be solved to optimality in split seconds by off-the-shelf solvers.

---

**Algorithm 1: Asynchronous Level Decomposition.**

---

1.　　Choose a gap tolerance $\text{tol}_\Delta$, upper bound $f_1^{\text{up}} > f_* + \text{tol}_\Delta$, lower bound $f_1^{\text{low}} < f_*$, $\alpha \in (0,1)$, $L > 0$, and $x_0$ a feasible point. Set $\mathbf{x}_1 = \hat{\mathbf{x}}_1 = \mathbf{x}_{\text{best}} = \mathbf{x}_0$, $\Delta_0 \leftarrow f_1^{\text{up}} - f_1^{\text{low}}$, $\hat{\Delta} \leftarrow \infty$, $\text{rr} \leftarrow 0$, $\mathcal{A} \leftarrow \{1, 2, \ldots, w\}$, $k \leftarrow 0$, $J^j \leftarrow \varnothing$ for $j \in \mathcal{A}$.

2.　　**for** $k \leftarrow 1$ **to** $k + 1$ **do**

3.　　**if** (5) does not hold and $\text{rr} = 0$ **then**

4.　　$\bar{\mathbf{x}}_k \leftarrow \mathbf{x}_k$, $\text{rr} \leftarrow w$, $\bar{f} \leftarrow \mathbf{c}^\mathsf{T} \bar{\mathbf{x}}_k$ and $\bar{g} \leftarrow \mathbf{c}$

5.　　**for all** $j \in \mathcal{A}$ **do**

6.　　**to_coordinate**[j] $\leftarrow$ *false* and

7.　　**coordinating**[j] $\leftarrow$ *true*

8.　　**end for**

9.　　**for all** $j \in \{1, \ldots, w\} \backslash \mathcal{A}$ **do**

10.　**to_coordinate**[j] $\leftarrow$ *true* and

11.　**coordinating**[j] $\leftarrow$ *false*

12.　**end for**

13.　**end if**

14.　Send $\mathbf{x}_k$ to all available workers $j \in \mathcal{A}$ and set $\mathcal{A} = \varnothing$

15.　Update the set $\mathcal{A}$ of idle workers and receive $\left(f^j(\mathbf{x}_{a(j)}), g^j_{a(j)}\right)$ from workers $j \in \mathcal{A}$

16.　Update $J^j \leftarrow J^j \cup \{a(j)\}$ for all $j \in \mathcal{A}$ and set $\mathcal{R} \leftarrow \varnothing$

17.　**for all** $j \in \mathcal{A}$ **do**

18.　**if coordinating**[j] = *true* **then**

19.　**coordinating**[j] $\leftarrow$ *false* and $\text{rr} \leftarrow \text{rr} - 1$

20.　$\bar{f} \leftarrow \bar{f} + f^j(\bar{\mathbf{x}}_{a(j)})$ and $\bar{g} \leftarrow \bar{g} + g^j_{a(j)}$

21.　**if** $\text{rr} = 0$ **then**

22.　Set $L \leftarrow \max\{L, \|\bar{g}\|\}$

23.　**if** $\bar{f} < f_k^{\text{up}}$ **then**

24.　$f_k^{\text{up}} \leftarrow \bar{f}$ and $\mathbf{x}_{\text{best}} \leftarrow \bar{\mathbf{x}}_k$

25.　**end if**

26.　**end if**

27.　**else**

28.　**if to_coordinate**[j] = *true* **then**

29.　Send $\bar{\mathbf{x}}_k$ to worker $j$ and set $\mathcal{R} \leftarrow \mathcal{R} \cup \{j\}$

30.　Set **to_coordinate**[j] $\leftarrow$ *false* and

31.　**coordinating**[j] $\leftarrow$ *true*

32.　**end if**

33.　**end if**

34.　**end for**

35.　Set $\mathcal{A} \leftarrow \mathcal{A} \backslash \mathcal{R}$

36.　Set $\Delta_k \leftarrow f_k^{\text{up}} - f_k^{\text{low}}$

37.　**if** $\Delta_k \leq \text{tol}_\Delta$ **then stop**: return $\mathbf{x}_{\text{best}}$ and $f_k^{\text{up}}$ **end if**

38.　**if** $\Delta_k \leq \alpha \hat{\Delta}$ **then** Set $\hat{\mathbf{x}}_k \leftarrow \mathbf{x}_{\text{best}}$ and $\hat{\Delta} \leftarrow \Delta_k$ **end if**

39.　$f_k^{\text{lev}} \leftarrow f_k^{\text{up}} - \alpha \Delta_k$

40.　**if** (4) is feasible **then**

41.　Get a new iterate $\mathbf{x}_{k+1}$ from the solution of (4)

42.　**else**

43.　Set $f_k^{\text{low}} \leftarrow f_k^{\text{lev}}$ and go to Step 36

44.　**end if**

45.　**if** $\mathbf{x}_{k+1}$ leads to infeasible subproblems **then**

46.　Add a feasibility cut to the MP (2) and go to Step 40

47.　**end if**

48.　Set $f_{k+1}^{\text{up}} \leftarrow f_k^{\text{up}}$, $f_{k+1}^{\text{low}} \leftarrow f_k^{\text{low}}$, $\hat{\mathbf{x}}_{k+1} \leftarrow \hat{\mathbf{x}}_k$ and $\bar{\mathbf{x}}_{k+1} \leftarrow \bar{\mathbf{x}}_k$

49.　**end for**

---

## 2.2. Convergence Analysis

To analyze the convergence of the mixed-integer asynchronous computing (ASYN) level decomposition (LD) described above, we rely as much as possible on Reference [31]. However, to account for the mixed-integer nature of the feasible set, we need novel developments like the ones in Theorem 3.1 below. Throughout this section, we assume $tol_\Delta = 0$, as well as the following:

**Hypothesis 1 (H1).** *all the workers are responsive;*

**Hypothesis 2 (H2).** *algorithm generates only finitely many feasibility cuts;*

**Hypothesis 3 (H3).** *the workers provide bounded subgradients.*

As for H1, the assumption H2 is a mild one: H2 holds, for instance, when $f$ is a polyhedral function, or when $\mathcal{X}$ has only finitely many points. The problem of interest satisfies both these properties, and, therefore, H2 is verified. Due to convexity of $f$, assumption H3 holds, e.g., if $\mathcal{X}$ is contained in an open convex set that is itself a subset of $Dom(f)$ (in this case, no feasibility cut will be generated). H3 also holds in our setting if subgradients are computed via basic optimal dual solutions of the second-stage subproblems. Under H3, we can ensure that the parameter $L$ in the algorithm is finite.

In our analysis, we use the fact that the sequences of the optimality gap, $\Delta_k$, and upper bound, $f_k^{up}$, are non-increasing by definition, and that the sequence of lower bound, $f_k^{low}$, is non-decreasing. More specifically, we update the lower bound only when the MP is infeasible. We count with $\ell$ the number of times the gap significantly decreases, meaning that the test of line 38 is triggered, and denote by $k(\ell)$ the corresponding iteration. We have the following by construction:

$$\Delta_{k(\ell+1)} \leq \alpha\Delta_{k(\ell)} \leq \alpha^2\Delta_{k(\ell-1)} \leq \cdots \leq \alpha^\ell\Delta_1 \quad \forall\, \ell = 1, 2, \ldots \tag{6}$$

As in Reference [31], $k(\ell)$ denotes a critical iteration, and $\mathbf{x}_{k(\ell)}$ denotes a critical iterate. We introduce the set of iterates between two consecutive critical iterates by $K^\ell := \{k(\ell)+1, \ldots, k(\ell+1)-1\}$. The proof of convergence of the ASYN LD consists in showing that the algorithm performs infinitely many critical iterations when $tol_\Delta = 0$. We start with the following lemma, which is a particular case of Reference [31], Lemma 3, and does not depend on the structure of $\mathcal{X}$.

**Lemma 1.** *Fix an arbitrary $\ell$ and let $K^\ell$ be defined as above. Then, for all $k \in K^\ell$, (a) the MP is feasible, and (b) the stability center is fixed: $\hat{\mathbf{x}}_k = \hat{\mathbf{x}}_{k(\ell)}$.*

Item (a) above ensures that the MP is well-defined and $f_k^{low}$ is fixed for all $k \in K^\ell$. Note that the lower bound is updated only when the MP is found infeasible, and this fact immediately triggers the test at line 38 of the algorithm. Similarly, Algorithm 1 guarantees that the stability center remains fixed for all $k \in K^\ell$, since an updated on the stability center would imply a new critical iteration.

**Theorem 1.** *Assume that $\mathcal{X}$ is a compact set and that H1-H3 hold. Let $tol_\Delta = 0$ in the algorithm, and then $\lim\limits_{k}\Delta_k = 0$.*

**Proof of Theorem 1.** By (6), we only need to show that the counter $\ell$ increases indefinitely (i.e., that there are infinitely many critical iterations). We obtain this by showing that, for any $\ell$, the set $K^\ell$ is finite; for this, suppose that $\Delta_k > \Delta > 0$ for all $k \in K^\ell$. We proceed in two steps, showing the following: (i) The number of asynchronous iterations between two consecutive coordination steps is finite, and (ii) the number of coordination steps in $K^\ell$ is finite, as well. If case (i) were not true, then (5) and Lemma 3.1(b) would give $\|\mathbf{x}_k - \hat{\mathbf{x}}_{k(\ell)}\|^2 \geq \|\mathbf{x}_{k-1} - \hat{\mathbf{x}}_{k(\ell)}\|^2 + \left(\frac{\alpha\Delta}{L}\right)^2$, for all $k \in K^\ell$ greater than the iteration $\bar{k}$ of the last coordination iterate. Applying this inequality recursively up to $\bar{k}$, we obtain $Diam(\mathcal{X})^2 \geq \|\mathbf{x}_k - \hat{\mathbf{x}}_{k(\ell)}\|^2 \geq (k - \bar{k} - 1)\left(\frac{\alpha\Delta}{L}\right)^2$. However, this inequality, together with H1 and $L < \infty$

(due to H3) contradicts the fact that $\mathcal{X}$ is bounded. Therefore, item (i) holds. We now turn our attention to the item (ii): Let $s, s' \in K^\ell$ such that $s < s'$ be the iteration indices of any two coordination steps. At the moment in which $\bar{\mathbf{x}}_{s'}$ is computed, the information $(f^j(\bar{\mathbf{x}}_s), g_s^j)$ is available at the MP for all $j = 1, \ldots, w$. As a result of the MP definition, the following constraints are satisfied by $\bar{x}_s$:

$$f^j(\bar{\mathbf{x}}_s) + \langle g_s^j, \bar{x}_{s'} - \bar{x}_s \rangle \le r^j \text{ and } \mathbf{c}^T \bar{x}_{s'} + \sum_{j=1}^{w} r^j \le f_{s'-1}^{\text{lev}}. \tag{7}$$

By assuming these inequalities and rearranging terms, we get $f(\bar{\mathbf{x}}_s) - f_{s'-1}^{\text{lev}} \le \langle \mathbf{c} + \sum_{j=1}^{w} g_s^j, \bar{\mathbf{x}}_s - \bar{\mathbf{x}}_{s'} \rangle \le \Gamma \|\bar{\mathbf{x}}_s - \bar{\mathbf{x}}_{s'}\|$, where the constant $\infty > \Gamma \ge L \ge \|\mathbf{c} + \sum_{j=1}^{w} g_s^j\|$ is ensured by H3. The definition of $f_{s'}^{\text{lev}} = f_{s'}^{up} - \alpha \Delta_{s'}$ and inequality $f(\bar{\mathbf{x}}_s) \ge f_{s'}^{up}$ gives $\|\bar{\mathbf{x}}_s - \bar{\mathbf{x}}_{s'}\| \ge \alpha \frac{\Delta_{s'}}{\Gamma} \ge \alpha \frac{\Delta}{\Gamma} > 0$. If there was an infinite number of coordination steps inside $K^\ell$, the compactness of $\mathcal{X}$ would allow us to extract a converging subsequence, and this would contradict the above inequality. The number of coordination steps inside $K^\ell$ is thus finite. As a conclusion of (i) and (ii), the index-set $K^\ell$ is hence finite, and the chain (6) concludes the proof. □

### 2.3. Dynamic Asynchronous Level Decomposition

In the asynchronous approach described in Algorithm 1, the component functions $f^j$ are statically assigned to workers—worker j always evaluates the same component function j. Likewise, the usual implementation of the synchronous LD strategy is to task workers with solving fixed sets of $Q_s$. We call these strategies static asynchronous LD and static synchronous LD. However, as previously mentioned, such task-allocation policies might result in significant idle times—even for the asynchronous method because we need the first-order information on all $f^j$ to compute valid bounds. To lessen the idle times, we implement dynamic-task-allocation strategies, in which component functions are dynamically assigned to workers as soon as they become available. Our dynamic allocation differs from Reference [15] because we do not use a list of iterates. To ease the understanding of the LD methods applied in this work—and to highlight their differences—we introduce a new figure: a coordinator process. The coordinator is responsible for tasking workers with functions to be evaluated. Note, however, that this additional figure is only strictly necessary in the dynamic asynchronous LD; in the other three methods, this responsibility can be taken by the master. Nonetheless, in all methods, the master has three roles: solving the MP, getting iterates, and requesting functions to be evaluated at the newly obtained iterates. By construction, in the synchronous methods, the master requests the coordinator to evaluate all functions $f^j$ at the same iterate, and it waits until the information of the all functions has been received to continue the process. On the other hand, in the asynchronous variants, the master computes a new iterate, requests the coordinator to evaluate it on possibly not all $f^j$, and receives information on outdate iterates from the coordinator. Given that the master has requested an iterate $\mathbf{x}'$ to be evaluated in some $f^j$, the main difference between the static and the dynamic asynchronous methods is that, in the static form, the coordinator always sends $\mathbf{x}'$ to the same worker who has been previously tasked with solving $f^j$, while in the dynamic one, the coordinator sends $\mathbf{x}'$ to any available worker.

## 3. Modeling Details and Case Studies

The general formulation of our SHTUC is presented in (8)–(19).

$$f_* = \min \sum_{g \in \mathcal{G}} \left[ \sum_{t \in \mathcal{T}} \left( \mathbf{CS}_g \cdot a_{gt} + \sum_{s \in \mathcal{S}} \mathbf{C}_g \cdot t g_{gts} \right) \right] + \sum_{b \in \mathcal{B}} \sum_{t \in \mathcal{T}} \mathbf{CL} \cdot (\delta_{bt}^+ + \delta_{bt}^-) + \sum_{s \in \mathcal{S}} f_s^\omega(v) \tag{8}$$

$$\text{s.t}: \sum_{o=t-\mathbf{TU}_g+1}^{t} a_{go} \leq I_{gt}, \quad \sum_{o=t-\mathbf{TD}_g+1}^{t} b_{go} \leq 1 - I_{gt} \tag{9}$$

$$a_{gt} - b_{gt} = I_{gt} - I_{gt-1}, z_{ht} - u_{ht} = w_{ht} - w_{ht-1} \tag{10}$$

$$z_{ht}, u_{ht}, w_{ht}, a_{gt}, b_{gt}, I_{gt} \in \{0, 1\} \tag{11}$$

$$I_{gt} \cdot \underline{\mathbf{P}}_g \leq tg_{gts} \leq I \cdot \overline{\mathbf{P}}_g \tag{12}$$

$$tg_{gts} - tg_{gt-1s} \leq I_{gt-1} \cdot \overline{\mathbf{R}}_g + (1 - I_{gt-1}) \cdot \mathbf{SU}_g \tag{13}$$

$$tg_{gt-1s} - tg_{gts} \leq I_{gt} \cdot \underline{\mathbf{R}}_g + (1 - I_{gt}) \cdot \mathbf{SD}_g \tag{14}$$

$$v_{hts} - v_{ht-1s} + f_{hts}^{\mathrm{v}}(q,s) + \mathbf{A}_{hts} = 0 \tag{15}$$

$$\underline{\mathbf{V}}_h \leq v_{hts} \leq \overline{\mathbf{V}}_h, w_{ht} \cdot \underline{\mathbf{Q}}_h \leq q_{hts} \leq w_{ht} \cdot \overline{\mathbf{Q}}_h, 0 \leq s_{hts} \leq \overline{\mathbf{S}}_h \tag{16}$$

$$0 \leq hg_{hts} \leq f_{hts}^{\mathrm{hg}}(q,s) \tag{17}$$

$$f_{bts}^{\mathrm{p}}(tg, hg, \delta^+, \delta^-) + \mathbf{WG}_{bts} - \mathbf{L}_{bt} = 0 \tag{18}$$

$$\underline{\mathbf{TL}}_l \leq f_{lts}^{\mathrm{l}}(tg, hg, \delta^+, \delta^-) \leq \overline{\mathbf{TL}}_l, \forall l \in \mathcal{L} \tag{19}$$

In our model, the indices and respective sets containing them are $g \in \mathcal{G}$ for thermal generators, $h \in \mathcal{H}$ for hydro plants, $b \in \mathcal{B}$ for buses, $l \in \mathcal{L}$ for transmission lines, and $t$ and $o \in \mathcal{T}$ for periods. In (5), thermal generators' start-up costs are $\mathbf{CS}$, and we assume that the shutdown cost is null. The thermal-generation costs are $\mathbf{C}$; $\mathbf{CL}$ is the per-unit cost of load shedding ($\delta^+$) and generation surplus ($\delta^-$). Expected future-operation cost for scenario $s$ is represented by the piecewise-affine function, $f_s^\omega(v^s) : \mathbb{R}^{|\mathcal{H}|} \to \mathbb{R}$, where $v^s \in \mathbb{R}^{|\mathcal{H}|}$ are the reservoir volumes in the last period of scenario $s$. The first-stage decisions are thermal generators' commitment, start-up, and shutdown, respectively, $I$, $a$, and $b$, and their hydro counterparts ($w$, $z$, and $u$). Set $\mathcal{X}$ in (1) contains the feasible commitments of thermal and hydro generators in our SHTUC, and it is defined by Constraints (9)–(11). In this work, we model the statuses of hydro plants with associated binary variables only in the first 48 h, to reduce the computational burden. For the remaining periods, the hydro plants are modeled only with continuous variables. The minimum up-time Constraint (9) ensures that, once turned on, thermal generator $g$ remains on for at least $\mathbf{TU}_g$ periods. Likewise, the minimum downtime in (9) requires that once $g$ has been turned off, it must remain off for at least $\mathbf{TD}_g$ periods. Constraints (10) guarantee the satisfaction of logical relations of status, start-up, and shutdown for thermal and hydro plants. The sets $\mathcal{Y}$s are defined by (12)–(19). Constraints (12) are the usual limits on thermal generation $tg$; (13) and (14) are the up and down ramp-rate limits, and the start-up and shutdown requirements of generators $g$. Equation (15) is the mass balance of the hydro plant $h$'s reservoir. The $\mathbf{A}_{hts}$ is the inflow to reservoir $h$ in period $t$ of scenario $s$. Moreover, the affine function $f_{hts}^{\mathrm{v}}(q, s) : \mathbb{R}^{2 \cdot |\mathcal{H}| \cdot |\mathcal{T}| \cdot |\mathcal{S}|} \to \mathbb{R}$ maps the inflow to $h$'s reservoir in period $t$ of scenario $s$ given the vectors of turbine discharge $q$ and spillage $s$. The constraints in (16) are the limits on reservoir volume, $v$, turbine discharge, $q$, and spillage, $s$. In (17), the piecewise-affine function $f_{hts}^{\mathrm{hg}}(q, s) : \mathbb{R}^{2 \cdot |\mathcal{H}| \cdot |\mathcal{T}| \cdot |\mathcal{S}|} \to \mathbb{R}$ bounds the hydropower generation $hg_{hts}$ of plant $h$. We use the classical DC network model: Equation (18) is the bus power balance, where the linear function $f_{bts}^{\mathrm{p}}(tg, hg, \delta^+, \delta^-) : \mathbb{R}^{|\mathcal{T}| \cdot |\mathcal{S}| \cdot (|\mathcal{G}| + |\mathcal{H}| + 2 \cdot |\mathcal{B}|)} \to \mathbb{R}$ maps the controlled generation at each bus into the power injection at bus $b$, $\mathbf{WG}_{bts}$ is the wind generation at bus $b$, and $\mathbf{L}_{bt}$ is the corresponding load at $b$. Lastly, (19) are the limits on the flow of transmission line $l$ in period $t$ and scenario $s$, defined by the affine function $f_{lst}^{\mathrm{l}}(tg, hg, \delta^+, \delta^-) : \mathbb{R}^{|\mathcal{T}| \cdot |\mathcal{S}| \cdot (|\mathcal{G}| + |\mathcal{H}| + 2 \cdot |\mathcal{B}|)} \to \mathbb{R}$.

We assess our algorithm on a 46-bus system with 11 thermal plants, 16 hydro plants, 3 wind farms, and 95 transmission lines. The system's installed capacity is 18,600 MW, from which 18.9% is due to thermal plants, hydro plants represent 68.1%, and wind farms have a share of 13%. We consider a one-week-long planning horizon with hourly discretization. Thus, a one-scenario instance of our

SHTUC would have 7848 binary variables and 5315 constraints at the first stage; and 36,457 continuous variables and 100,949 constraints for each scenario in the second stage. Furthermore, the weekly peak load in the baseline case is 11,204 MW—nearly 60.2% of the installed capacity. The hydro plants are distributed over two basins and include both run-of-river ones and plants with reservoirs capable of regularization. Further information about the system can be found in the multimedia files attached.

The uncertainty comes from wind generation and the inflows. In all tests, we use a scenario set with 256 scenarios. To assess how our algorithm performs in distinct scenario sets, three sets (A, B, and C) are considered. Moreover, we use three initial useful-reservoir-volume levels: 40%, 50%, and 70%. The impact of different load levels on the performance of our algorithms is analyzed through three load levels: low (L), moderate (M), and high (H). Level H is our baseline case regarding load. Levels M and L have the same load profile as H's, but with all loads multiplied by factors of 0.9 and 0.8, respectively. Lastly, to investigate how our algorithm's convergence rate is affected by different choices of initial stability centers, we implement two strategies for obtaining the initial stability center. In both strategies, we solve an expected-value problem, as defined in Reference [5]. In the first one, we use the classical Benders decomposition (BD) with a coarse relative-optimality-gap tolerance of 10% to get a, possibly, low-quality stability center (LQSC). To obtain the stability center of hopefully high quality, which we refer to as high-quality stability center (HQSC), we solve the expected-value problem directly with Gurobi 8.1.1 [33] with a relative-optimality-gap tolerance of 1%. The time limit for obtaining the initial stability centers LQSC and HQSC is set to 5 min. Additionally, the computing setting consists of seven machines of two types: 4 of them have 128 GB of RAM and two Xeon E5-2660 v3 processors with 10 cores clocking at 2.6 GHz; the other 3 machines have 32 GB of RAM and two Xeon X5690 processors with cores cores clocking at 3.47 GHz. All machines are in a LAN with 1-Gbps network interfaces. We test two machine combinations. In the first one, in Combination 1, there are four 20-core machines and one with 12 cores. In Combination 2, we replace one machine with 20 cores by 2 with 12 cores. Regardless of the combination, one 12-core machine is defined as the head node, where only the master is launched. Except for the master—for which Gurobi can take up to 10 cores—for all other processes, i.e., the workers, Gurobi is limited to computing on a single core.

Our computing setting is composed of machines with different configurations. Naturally, solving the same component function in two distinct machines may result in different outputs—and different runtimes. Consequently, the path taken by the MP across iterations might change significantly between experiments on the same data. More specifically to asynchronous methods, the varying order of information arrival to the MP may also yield different convergence rates. Hence, to reduce the effect of these seemingly random behaviors, we conducted 5 experiments for each problem instance. Therefore, our testbed $\mathcal{E}$ is defined as $\mathcal{E} = \{40, 50, 70\} \times \{A, B, C\} \times \{L, M, H\} \times \{$LQ-SC, HQ-SC$\} \times \{$Trial 1, ... , Trial 5$\} \times \{$Combination 1, Combination 2$\}$—we have 54 problems and 540 experiments. In all instances in $\mathcal{E}$, we divide $\mathcal{S}$ into 16 subsets. Thus, following our previous definitions, w = 16 and any subset $\mathcal{P}_j$ is such that $|\mathcal{P}_j| = 16$. Additionally, we set a relative-optimality-gap tolerance of 1% and a time limit of 30 min for all instances in $\mathcal{E}$. Gurobi 8.1.1 is used to solve the MILP MP and the component functions (linear-programming problems) that form the subproblem. The inter-process communication is implemented with mpi4py and Microsoft MPI v10.0.

## 4. Results

In this section, the methods are analyzed based on their computing-time performances. We focus on this metric because our results have not shown significant differences among the methods for other metrics, e.g., optimality gap and upper bounds. In addition to analyzing averages of the metric, we use the well-known performance profile [34]. Multimedia files containing the main results for the set $\mathcal{E}$ are attached to this work.

Figure 1 presents the performance profiles of the methods considering the experiments $\mathcal{E}$. In Figure 1, $\rho(\tau)$ and $\tau$ are, respectively, the probability that the performance ratio of a given method is within a factor $\tau$ of the best ratio, as in Reference [34]. Applying the classical Benders decomposition

(BD) on the set {40, 50, 70} × {A} × {L, M, H} × {Combination 1} results in the convergence only of the problem in {70} × {A} × {M} × {Combination 1}, for which BD converges to a 1%-optimal solution in 1281.42 s. Thus, it is reasonable to expect that the classical BD would also perform poorly for the remaining experiments $\mathcal{E}$.

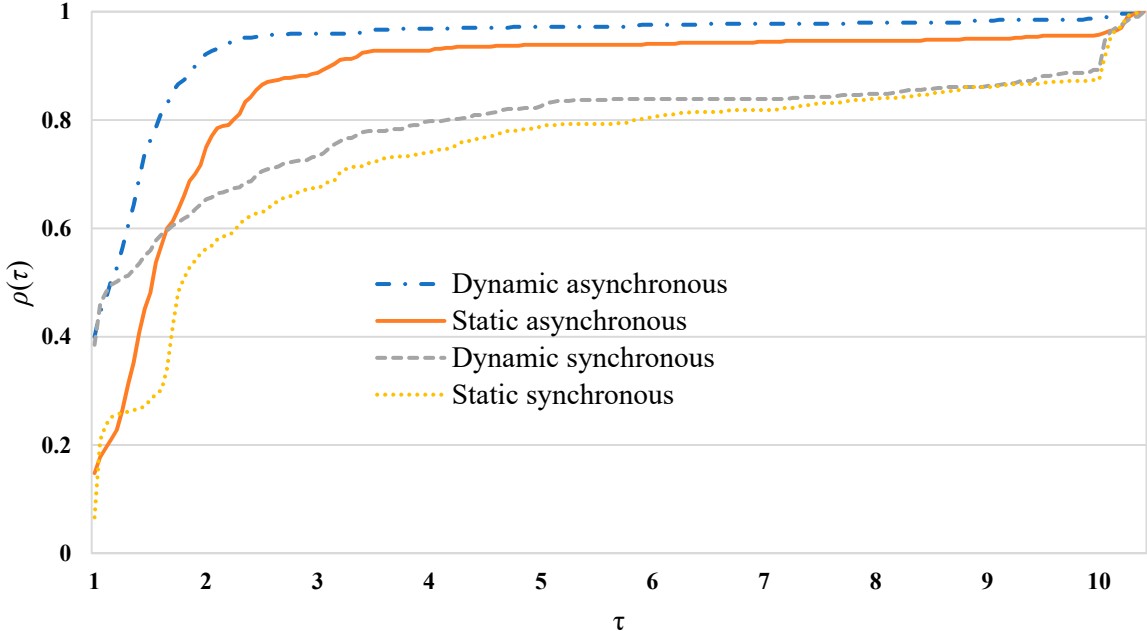

**Figure 1.** Performance profiles over the set $\mathcal{E}$.

In Figure 1, we see that the dynamic asynchronous LD outperforms all other methods for most instances $\mathcal{E}$. Its performance ratio is within a factor of 2 from the best ratio for about 500 instances (about 92% of the total). Moreover, the static asynchronous LD has a reasonable overall performance—it is within a factor of 2 from the best ratio for more than 400 instances. Moreover, we see that the dynamic-allocation strategy provides significant improvements for both the asynchronous and synchronous LD approaches. The dynamic synchronous LD converges faster than its static counterpart for most of the experiments. Figures 2 and 3 show the performance profiles considering only instances in $\mathcal{E}$ with machine Combinations 1 and 2, respectively.

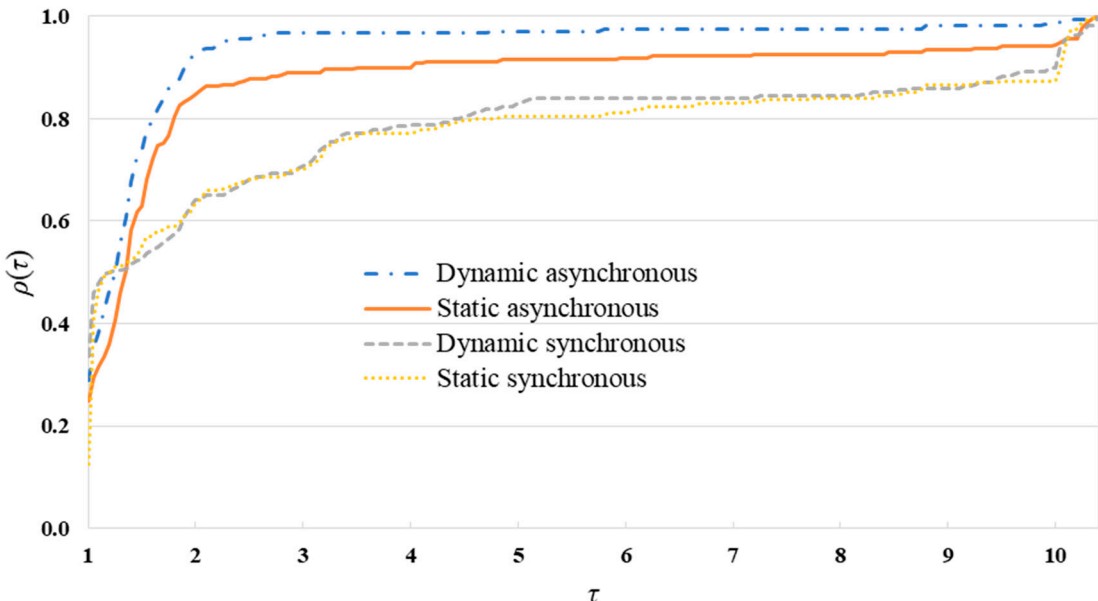

**Figure 2.** Performance profiles for the instances with machine Combination 1.

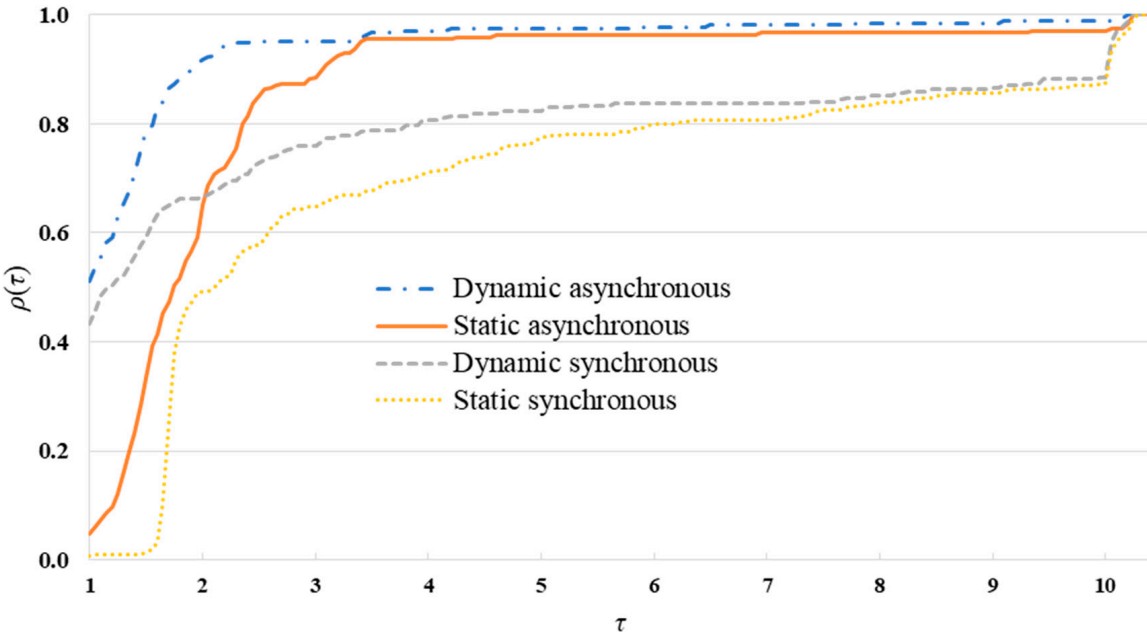

**Figure 3.** Performance profiles for the instances with machine Combination 2.

Figure 2 illustrates that, for a distributed setting in which workers are deployed on machines with identical characteristics, the performances of the methods with dynamic allocation and those with static allocation are similar. Nonetheless, we see that the asynchronous methods still outperform the synchronous LD for most experiments.

In contrast to Figure 2, Figure 3 shows that the dynamic-allocation strategy provides significant time savings for the instances in $\mathcal{E}$ with Machine Combination 2. This is due to the great imbalance between the different machines in Combination 2—machines with processors Xeon E5-2660 v3 are much faster than those with processors Xeon X5690.

Table 1 gives the average wall-clock computing times over subsets of $\mathcal{E}$. From this table, we see that the relative average speed-up of the dynamic and static asynchronous LD over the entire set $\mathcal{E}$ w.r.t. The static synchronous LD are 54% and 29%, respectively—considering the dynamic synchronous LD, the speed-ups are 45% and 16%, respectively. Moreover, we see that the time savings are more significant for harder-to-solve instances, e.g., instances with high load and/or low-quality initial stability centers. Additionally, Table 1 shows that the dynamic asynchronous LD provides considerable reductions in the standard deviations of the elapsed computing times, in comparison with the other methods. For example, for the problems with high load level (H), the dynamic asynchronous LD has a standard deviation of about 16%, 13%, and 27% smaller than that of the static asynchronous LD, dynamic synchronous LD, and static synchronous LD, respectively.

Based on the data from Table 1, we can compute the speed-up provided by our proposed dynamic ASYN LD w.r.t., and the other three variants are considered here. To better appreciate such speed-ups, we show them in Table 2, where we see that the proposed ASYN LD provides consistent speed-ups over the entire range of operating conditions considered here.

The advantages of the asynchronous methods are made clearer in Figure 4, where we see that not only the asynchronous methods provide (on average) better running times but also present significantly less variation among the problems in $\mathcal{E}$. The latter is relevant in the day-to-day operations of ISOs, since, if there are stochastic hydrothermal unit-commitment (SHTUC) cases that take significantly more time to be solved than the expected, subsequent operation steps that depend on the results of the SHTUC might be affected. Take, for instance, the case from the Midcontinent Independent System Operator reported in Reference [3], where the (deterministic) UC is reported to have solution times varying from just 50 to over 3600 s. Such variation can be problematic in the day-to-day operation of

power systems since it may disrupt tightly scheduled operations. Naturally, methods that can reduce such variance and still produce high-quality solutions in reasonable times are appealing.

**Table 1.** Average elapsed time and standard deviation in seconds.

|  | Asynchronous | | Synchronous | |
|---|---|---|---|---|
|  | **Dynamic** | **Static** | **Dynamic** | **Static** |
| 40 | 135 (340) | 240 (497) | 364 (569) | 338 (490) |
| 50 | 130 (279) | 222 (414) | 195 (293) | 267 (404) |
| 70 | 127 (249) | 139 (242) | 161 (157) | 250 (300) |
| A | 143 (280) | 226 (418) | 187 (270) | 216 (303) |
| B | 137 (340) | 192 (406) | 305 (515) | 347 (530) |
| C | 112 (247) | 184 (377) | 229 (335) | 291 (340) |
| L | 102 (195) | 157 (317) | 172 (398) | 201 (396) |
| M | 108 (179) | 112 (111) | 142 (126) | 205 (210) |
| H | 182 (425) | 333 (585) | 405 (492) | 449 (506) |
| HQSC | 105 (202) | 129 (193) | 156 (258) | 159 (118) |
| LQSC | 156 (357) | 272 (523) | 324 (473) | 411 (534) |
| Combination 1 | 120 (282) | 207 (447) | 227 (387) | 243 (416) |
| Combination 2 | 141 (301) | 194 (348) | 253 (393) | 327 (393) |

The rows indicate that the average elapsed times and standard deviation given in parentheses are computed considering only the instances in $\mathcal{E}$ with the parameter given in the column 1. For example, the averages and respective standard deviations in row 3 are computed considering all experiments for which the initial useful-reservoir-volume level is 40%. Likewise, rows 4 and 7 provide the averages over instances with scenario set A and load level L, respectively. In rows 10 and 11, HQSC and LQSC stand for high-quality stability center and low-quality stability center, respectively.

**Table 2.** Speed-ups in % provided by the asynchronous computing (ASYN) with respect to the level decomposition (LD)

|  | **Static SYN LD** | **Dynamic SYN LD** | **Static ASYN LD** |
|---|---|---|---|
| 40 | 60 | 63 | 44 |
| 50 | 51 | 33 | 41 |
| 70 | 49 | 21 | 9 |
| A | 34 | 23 | 37 |
| B | 61 | 55 | 29 |
| C | 61 | 51 | 39 |
| L | 49 | 41 | 35 |
| M | 47 | 24 | 4 |
| H | 59 | 55 | 45 |
| HQSC | 34 | 33 | 19 |
| LQSC | 62 | 52 | 42 |
| Combination 1 | 50 | 47 | 42 |
| Combination 2 | 57 | 44 | 27 |

As in Table 1, the rows indicate that the average speed-up computed considering only the instances in $\mathcal{E}$ with the parameter given in the column 1. Moreover, the columns indicate the method the speed-up is computed for. For example, column Static SYN (synchronous computing) LD gives the speed-ups provided by the ASYN LD over instances in the first column w.r.t. to the static synchronous level decomposition.

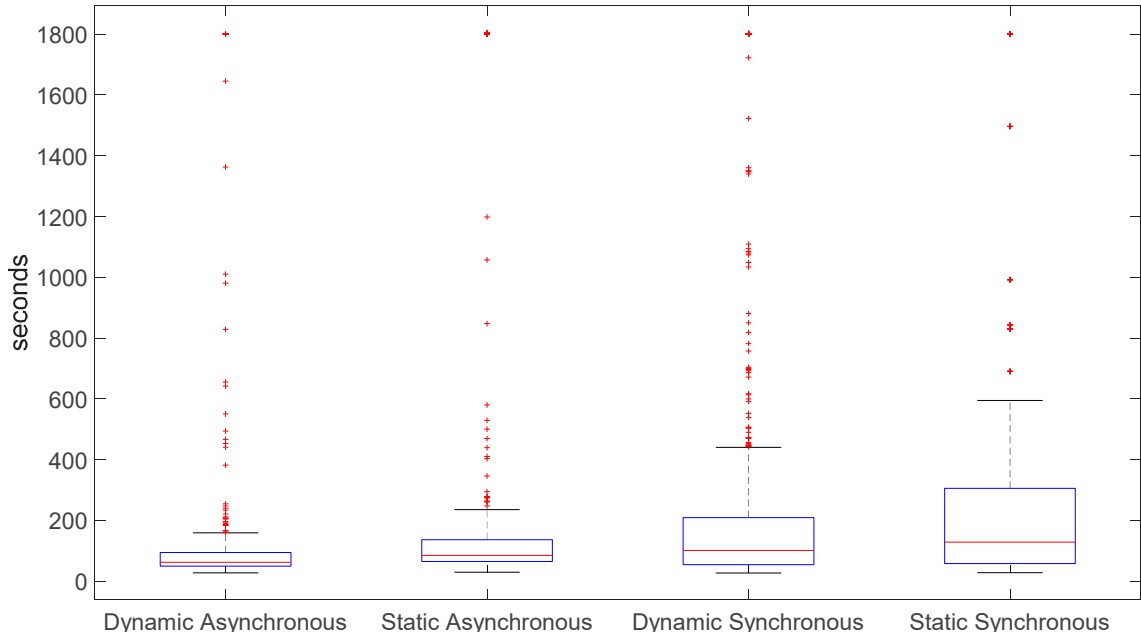

**Figure 4.** Boxplot of the methods over the set $\mathcal{E}$.

## 5. Conclusions

In this work, we present an extension of the asynchronous level decomposition of Reference [31] in a Benders-decomposition framework. We show a convergence analysis of our algorithm, proving that it converges to an optimal solution, if one exists, in finite-many iterations. Our experiments are conducted on an extensive testbed from a real-life-size system. The results show that the proposed asynchronous algorithm outperforms its synchronous counterpart in most of the problems and provides significant time savings. Moreover, we show that the improvements provided by the asynchronous methods over the synchronous ones are even more evident in a distributed-computing setting with machines of different computational powers. Additionally, we show that the asynchronous method is further enhanced by implementing a dynamic-task-allocation strategy.

**Author Contributions:** Conceptualization, B.C., E.C.F., and W.d.O.; methodology, B.C., E.C.F., and W.d.O.; software, B.C.; validation, B.C.; formal analysis, E.C.F. and W.d.O.; investigation, B.C., E.C.F., and W.d.O.; resources, B.C. and E.C.F.; data curation, B.C.; writing—original draft preparation, B.C., E.C.F., and W.d.O.; writing—review and editing, B.C., E.C.F., and W.d.O.; visualization, B.C., E.C.F., and W.d.O.; supervision, E.C.F. and W.d.O. All authors have read and agreed to the published version of the manuscript.

**Funding:** The third author acknowledges financial support from the Gaspard-Monge program for Optimization and Operations Research (PGMO) project "Models for planning energy investment under uncertainty".

**Conflicts of Interest:** The authors declare no conflict of interest.

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
