# Peer review of "A Mixed-Integer and Asynchronous Level Decomposition with Application to the Stochastic Hydrothermal Unit-Commitment Problem"

_algorithms, doi:10.3390/a13090235_

Round 1
Reviewer 1 Report
- Rewrite the Introduction. The inspiration of your work must be highlighted in Introduction.
- In general, all variables and Greek letters should be in italic format, and all constants should not be in italic format. Vectors or matrix variables should be in bold and italic format. Please double check the equations used in the manuscript.
- The rule for acronyms is to first spell out the reference, followed by the acronym in parentheses, at its first use. Refer to the acronym thereafter, but repeat the full spelling and acronym at its first use in any major section. You have not yet defined this one.
- I suggest the authors number all the equations used in the paper;
- Re-edit equation (3);
- There are some weird symbols in the paper, the authors must update them;
- Format the “Table 1. Average elapsed time and standard deviation in seconds” with three lines;
- More experiments and analyses must be provided.
- Except the methods used in the paper, some of the most representative computational intelligence algorithms can be used to solve the problems, like monarch butterfly optimization (MBO), earthworm optimization algorithm (EWA), elephant herding optimization (EHO), moth search (MS) algorithm. these must be clearly pointed out in Section Conclusion.
Based on my above comments, I proposed that the paper is accepted with major revision.
Author Response
Dear referee,
Firstly, we would like to express our sincere gratitude for your thorough review of our manuscript. We believe that your suggestions have improved the overall quality of our work, and we hope that the changes made to the manuscript are satisfactory and successfully answer all of your questions.
To ease the understanding of the modifications we have made, we have separated your suggestions (in red) into appropriate pieces and responded to each of them separately, as follows.
Rewrite the Introduction. The inspiration of your work must be highlighted in Introduction.
Based on your suggestion, we have updated the introduction to better highlight the difficulty of solving the unit-commitment problem even in its deterministic form. To that end, we have now included two recent works (both from 2020) that report time limits for solving the deterministic UC in Brazil (ref [2]), and in a system operator in the USA (ref [3]). This new information is located on lines 35-40 of the revised manuscript. We hope that such information gives a better appreciation of the problem’s difficulty since in its uncertain form, which is the one we tackled, the UC is even more challenging. Finally, this is precisely the reasoning what we use on lines 41-46 of the updated manuscript to motivate the need for efficient solution methods for the uncertain UC, such as our proposed method.
A minor change has been made on lines 47-48 to accommodate the aforementioned changes in the introduction, namely, the text goes from “To model uncertainty arising from these sources” to “In particular, to model the uncertainty arising from renewable sources”.
In general, all variables and Greek letters should be in italic format, and all constants should not be in italic format. Vectors or matrix variables should be in bold and italic format. Please double check the equations used in the manuscript.
We have checked and corrected the document as requested. An example of such change can be found in equation 1.
The rule for acronyms is to first spell out the reference, followed by the acronym in parentheses, at its first use. Refer to the acronym thereafter, but repeat the full spelling and acronym at its first use in any major section. You have not yet defined this one.
We have followed your recommendation and appropriately defined the following acronyms: ISO, UC, MISO, ASYN, SYN, BD, LD, and MP. The definitions are given in every major section that these acronyms appear.
I suggest the authors number all the equations used in the paper.
As suggested, we have numbered the equations that are not inline. All other numbered equations have been updated accordingly.
Re-edit equation (3);There are some weird symbols in the paper, the authors must update them;
We believe this might be due to (Microsoft Word) compatibility issues. We would like to hear back from you if such problems persists. A pdf version of our manuscript is available at the link https://bit.ly/2D4if4U.
Format the “Table 1. Average elapsed time and standard deviation in seconds” with three lines;
As suggested, we have included lines to better separate the rows in Table 1 (now located at line 480 of the manuscript). However, our impression is that four lines (two on the top and bottom of the table, and two separating the headers) instead of three seem to be aesthetically better.
More experiments and analyses must be provided.
Although, as authors, we would also like to see how our proposed method performs over even larger sets of experiments than the ones we have used, it is our understanding that we have used a sizable dataset for our tests: there are 54 problems and 540 experiments. The problems are variations of operating conditions which are used to demonstrate how the stochastic unit commitment can present significant variations of solution times under different conditions. As for the experiments, they are variations of the set of machines used for solving the problems as well as several runs of the same problem with the same machines to provide statistical relevance. We also feel that more problems and experiments could cause more harm than good: they may not bring new insights and could cause the text to become too dense and hard to follow. Nevertheless, we are open to new ideas of problems that you think may be relevant to the method and context of our work. As for the analyses, our work is mainly concern with solution times. To better show this and also to better appreciate the benefits of our proposition, we have now inserted a new table, Table 2 on line 487, along with a paragraph on lines 483-486, to showcase the speed-ups provided by the dynamic asynchronous level decomposition with respect to the other three variants of the level decomposition that we consider in this work.
Except the methods used in the paper, some of the most representative computational intelligence algorithms can be used to solve the problems, like monarch butterfly optimization (MBO), earthworm optimization algorithm (EWA), elephant herding optimization (EHO), moth search (MS) algorithm. these must be clearly pointed out in Section Conclusion.
We appreciate this reviewer's comment on these methods as they were entirely missing from our work. Nonetheless, our main interest in this manuscript is in mathematical-programming methods. Therefore, we feel that including the methods mentioned above in the conclusion would seem out of place. Thus, we have instead included a paragraph in the introduction dedicated to the reasoning behind our choice for mathematical-programming methods, as well as to point out the importance of computational intelligence algorithms — where we kindly borrow the mentioned methods to use them as examples. Furthermore, we have also included a reference ([33]) which reviews heuristic and metaheuristic methods applied to unit-commitment problems. This new paragraph can be found on lines 132-141 of the updated manuscript.
Reviewer 2 Report
This paper proposes an asynchronous level decomposition method to solve stochastic hydrothermal unit-commitment (UC) problems with mixed-integer variables in the first stage.
The main problem that this paper tries to solve is the load imbalance issues in the stochastic hydrothermal UC problems, which are caused by the inherent run-time differences amongst the subproblem's optimization models, unequal equipment, communication overheads, etc.
The proposed asynchronous level decomposition methods are not new in that the proposed solutions are built on top of the existing previous work [29][28][13].
By combining and extending multiple existing techniques, however, the proposed dynamic asynchronous algorithm could outperform existing asynchronous and synchronous algorithms, as demonstrated in the evaluation section.
Providing the convergence analysis of the proposed methods is also another main contribution of this work, since most of previous work relies on the convexity of the problems, which is not applicable to the mixed-integer problems that this paper deals with.
The convergence analysis part of this paper is hard to understand, and there are several minor grammatical/presentation errors that need to be fixed.
However, the performance evaluation using two different computing platforms seems to be well constructed and nicely demonstrate the performance benefits of the proposed dynamic asynchronous algorithm.
Therefore, this paper seems to have enough contributions to be published to this journal.
Author Response
Dear referee,
Thank you for your kind review of our manuscript. We are glad that your general opinion on our work points to its acceptance in Algorithms. As for your recommendation of double checking the grammar, we have used a software to look for grammatical errors, but we have not been able to find any. Nonetheless, it is certainly possible that we still have overlooked something, but, at this point, we are unable to find the grammatical errors that you have kindly pointed out in your review. As for the presentation, we have improved Tables 1 and 2 in the manuscript by adding lines to better separate their rows. Moreover, we have also improved the notation of variables and constants to ease the understanding of the mathematical developments. Such modifications, unfortunately, cannot be seen through the Word’s track-change tool. But, as an example, the changes can be seen in Equation 1.
Finally, we want to mention that the convergence analysis of our algorithm is indeed not trivial: this a known drawback of bundle methods. However, the presented mathematical developments mirror as much as possible those of references [30] and [31].
Round 2
Reviewer 1 Report
The revised paper is resubmitted to “algorithms”. This version is little better than before. Few comments have been considered, so many problems exist in this revised version. My detailed comments are below:
- Rewrite Abstract. Your point is your own work that should be further highlighted.
- The usage of acronyms is so disorder;
- I strongly suggest the authors re-edit all the equations;
- When reviewing the related work, some of the most presentative work must be included, like Meta-heuristic framework: Quantum inspired binary grey wolf optimizer for unit commitment problem, Computers & Electrical Engineering, Volume 70, August 2018, Pages 243-260.
- The mathematical symbols are used in disorder;
- There are weird symbols in Eq. (5);
- All the figures must be further refined;
- The Section Conclusions is missed. The structure of the paper is incomplete.
- Except the methods used in the paper, some of the most representative computational intelligence algorithms can be used to solve the problems, like monarch butterfly optimization (MBO), earthworm optimization algorithm (EWA), elephant herding optimization (EHO), moth search (MS) algorithm. these must be clearly pointed out in Section Conclusion.
Based on my above comments, I proposed that the paper is accepted with major revision.
Author Response
Dear reviewer,
Once again, we appreciate your constructive criticism about our work. For your convenience, we answer your comments point-by-point in the following.
Rewrite Abstract. Your point is your own work that should be further highlighted.
A: We have highlighted that the asynchronous level decomposition in a mixed-integer context is an innovation. We also point out that the combination of this novel algorithm with appropriate task allocation is also a novelty in its own right and it far outperforms the state-of-the-art.
The usage of acronyms is so disorder;
A: We have revised the text and we feel that the acronyms are appropriately placed according to the guidelines given by this journal. However, if it is your understanding that we have interpreted the guidelines incorrectly, or if you have found a mistake of any sort regarding the acronyms, please, let us know.
I strongly suggest the authors re-edit all the equations;
A: Most equations have been edited to more consistently reflect the following rules: variables are shown in italic, vectors and matrices of constants are given in non-italic and boldface, subscripts are non-italic.
When reviewing the related work, some of the most presentative work must be included, like Meta-heuristic framework: Quantum inspired binary grey wolf optimizer for unit commitment problem, Computers & Electrical Engineering, Volume 70, August 2018, Pages 243-260.
A: Regarding the evolutionary algorithms, we have followed the suggestions of the editor.
The mathematical symbols are used in disorder;
A: We cannot say we fully understand what this reviewer means by disorder in this context. The definition of the mathematical elements are given either immediately before or after their respective first uses. If there is any error in this regard, please, let us know.
There are weird symbols in Eq. (5);
A: We believe such weird symbols are due to errors in conversion.
All the figures must be further refined;
A: It is not clear what this reviewer means by "refined".
The Section Conclusions is missed. The structure of the paper is incomplete.
A: The then Discussion section has been transformed into a Conclusion. We opted for this because enough discussion has been made while presenting the results.
Except the methods used in the paper, some of the most representative computational intelligence algorithms can be used to solve the problems, like monarch butterfly optimization (MBO), earthworm optimization algorithm (EWA), elephant herding optimization (EHO), moth search (MS) algorithm. these must be clearly pointed out in Section Conclusion.
A: Regarding the evolutionary algorithms, we have followed the suggestions of the editor.